# Q-VLM: Post-training Quantization for Large Vision-Language Models

**Changyuan Wang[1], Ziwei Wang[3], Xiuwei Xu[2], Yansong Tang[1*], Jie Zhou[2], Jiwen Lu[2]**

[1]Shenzhen International Graduate School, Tsinghua University, China
[2]Department of Automation, Tsinghua University, China
[3]School of Electrical and Electronic Engineering, Nanyang Technological University
{wangchan22@mails.,xxw21@mails.,tang.yansong@sz.}tsinghua.edu.cn;
{jzhou@,lujiwen@}tsinghua.edu.cn; ziwei.wang@ntu.edu.sg

## Abstract

In this paper, we propose a post-training quantization framework of large vision-language models (LVLMs) for efficient multi-modal inference. Conventional quantization methods sequentially search the layer-wise rounding functions by minimizing activation discretization errors, which fails to acquire optimal quantization strategy without considering cross-layer dependency. On the contrary, we mine the cross-layer dependency that significantly influences discretization errors of the entire vision-language model, and embed this dependency into optimal quantization strategy searching with low search cost. Specifically, we observe the strong correlation between the activation entropy and the cross-layer dependency concerning output discretization errors. Therefore, we employ the entropy as the proxy to partition blocks optimally, which aims to achieve satisfying trade-offs between discretization errors and the search cost. Moreover, we optimize the visual encoder to disentangle the cross-layer dependency for fine-grained decomposition of search space, so that the search cost is further reduced without harming the quantization accuracy. Experimental results demonstrate that our method compresses the memory by 2.78x and increase generate speed by 1.44x about 13B LLaVA model without performance degradation on diverse multi-modal reasoning tasks.[1]

## 1 Introduction

Large vision-language models (LVLMs) [31, 53] have achieved outstanding performance in a large number of multi-modal reasoning tasks such as visual question answering [45, 28], embodied instruction following [1] and robot navigation [2, 16], which are benefited from numerous network parameters and vast training data. Despite of the high accuracy and generalization ability across different tasks, the extreme computational cost hinders the deployment on resource-limited mobile devices in wide realistic deployment scenarios. Moreover, LVLMs sequentially generate the response with multiple forward passes, which further increases the computation burden to accomplish the task. Therefore, it is highly demanded to reduce the model complexity of LVLMs in practical deployment.

To reduce the model complexity, model compression techniques have been presented to accelerate computation and save the storage space including pruning [18, 51], quantization [19, 13, 23], low-rank decomposition [26, 20] and efficient architecture design [38, 17]. Among these methods, quantization replaces the float numbers with quantized ones and substitutes multiply-accumulate (MAC) operations with integer arithmetic for significant efficiency enhancement. Due to the intractability of the training data and the unbearable training cost of LVLMs, post-training quantization [33, 36, 48] is leveraged

---

[1]Code is available at https://github.com/ChangyuanWang17/QVLM

to reduce bitwidths of weights and activations, which only searches rounding functions with a small calibration set with frozen network parameters. Searching rounding functions that minimize model prediction errors causes extremely high search cost due to the large space, and conventional methods [12, 29] sequentially search the layer-wise rounding functions by minimizing the activation discretization errors. However, ignoring cross-layer dependency of discretization errors fails to acquire the optimal rounding strategy and degrades the performance significantly.

In this paper, we present an accurate post-training quantization framework called Q-VLM to accelerate large vision-language models for efficient multi-modal reasoning. Different from existing methods which sequentially search the layer-wise rounding functions, we mine the cross-layer dependency of output discretization errors across layers, and employ the dependency to efficiently search the optimal rounding functions that minimize the quantization noise of the entire model. More specifically, we observe the significant correlation between the activation entropy and the discretization error dependency with the following layers. We then employ the entropy as the proxy to decompose the large search space from the entire model to smaller blocks containing multiple layers, and rounding functions are searched with the goal of minimizing the block-wise discretization errors. Therefore, the quantized model remains competitive performance with original full-precision counterparts with trivial additional search cost. Moreover, we optimize the visual encoder to disentangle the cross-layer dependency for fine-grained search space decomposition, so that precise rounding functions can be acquired with further reduced search cost. Our Q-VLM can still generate plausible response in multi-modal reasoning with 4-bit quantization because of the precise rounding functions, and compresses the memory by 2.78x and increase the generate speed by 1.44x about 13B LLaVA model. We evaluate our method with the LLaVA and the MoE-LLaVA models in different bitwidth settings, and the results in various visual question answering datasets demonstrate that our Q-VLM outperforms the state-of-the-art post-training methods significantly with negligible search overhead.

## 2 Related Work

### 2.1 Large Vision-language Model

Large vision-language models (LVLMs) have achieved remarkable performance because of the fast adaptation on different downstream tasks with high generalization ability, which benefits from large-scale image-text pairs [39, 21] and strong generalization capabilities of pre-trained large language models (LLMs) [4, 40]. The instruction-following ability and multi-modal representations extracted by LVLMs are general across tasks, which are usually applied in a wide variety of multi-modal reasoning tasks such as visual question answering [45, 28], embodied instruction following [1] and robot navigation [2, 16]. Early attempts introduced rich commonsense in LLMs to vision-language representation learning, which effectively exploits LLMs by treating visual inputs as conditional information. In particular, BLIP [25, 24] leveraged data filtering techniques to enhance performance in tasks such as visual question answering (VQA) and image captioning. While these models exhibited extraordinary vision-language reasoning capabilities, their zero-shot abilities were limited due to the absence of explicit instruction during training. Recent studies including LLaVA [31] and InstructBLIP [30] aim to enhance LVLMs' zero-shot capabilities by aligning them more closely with human preferences. They finetuned LVLMs with visual instruction samples where the models were required to complete the human instruction according to the visual information. Despite the notable performance gains from the large model sizes, the computational complexity and the storage cost prohibit LVLMs from being deployed in resource-limited devices for realistic deployment. Lightweight LVLMs such as TinyGPT-V[49] and TinyLLaVA [53] endeavors explore the extensive domain of large multimodal models focusing on leveraging small-scale models and achieve efficient LVLMs architecture designs. MoE-LLaVA [28] constructs a spare MoE-based model, which identifies a sparse pathway by simultaneously handling image and text features to achieve comparable performance with fewer activated parameters. However, the model inference cost still exceeds the resource budget of mobile devices or robots because of the low compression ratio.

### 2.2 Post-training Quantization

Network quantization substitutes full-precision tensors with low-precision values and replaces multiply-accumulate operations with integer arithmetics, which significantly reduces the storage and computational cost of neural networks. Traditional quantization-aware training (QAT) meth-

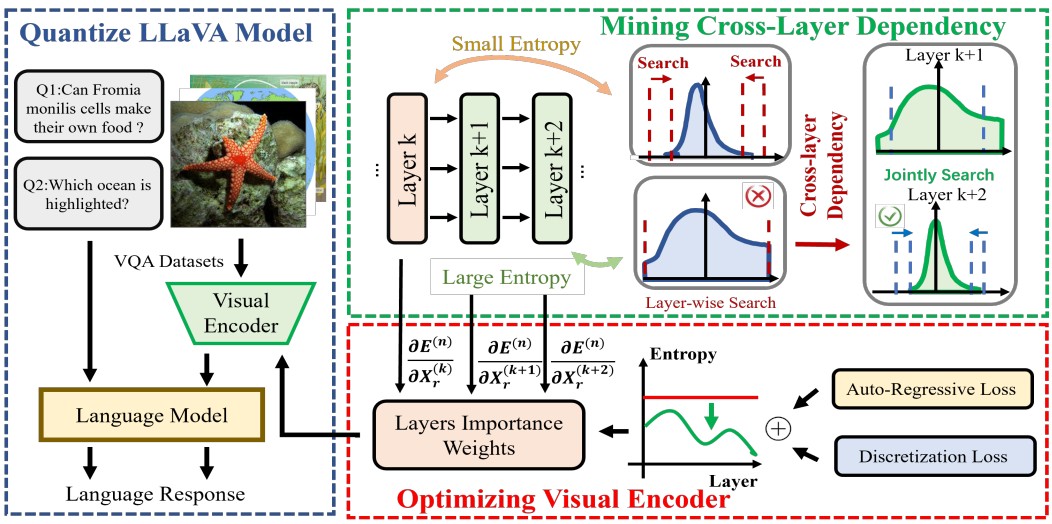

Figure 1: The overall pipeline of our method. We employ entropy as the proxy to represent cross-layer dependency for efficient block assignment, which decomposes the large search space from the entire model to blocks containing multiple layers. Moreover, the visual encoder is further optimized for fine-grained search space decomposition.

ods [6, 32] need to finetune network weights with the full training set for rounding, which is less practical since the data and resources for training may not be accessible for most users. Recently, post-training quantization (PTQ) [11, 33, 44, 10, 42, 41] has aroused extensive interest, which leverages a small calibration set to search for the optimal threshold in rounding functions with significantly reduced data demand and optimization cost. Choukroun *et al.* [7] minimized the $l_2$ distance between quantized and full-precision tensors to mitigate evident task performance degradation, while Zhao *et al.* [52] duplicated channels with outliers and halved their values to reduce clipping loss without amplifying rounding errors. Liu *et al.* [33] preserved relative ranking orders of self-attention in vision transformers to mitigate information loss during post-training quantization and explored a mixed-precision quantization strategy based on the nuclear norm of attention maps and features. Zero-shot PTQ further extends the boundaries for efficiently quantizing neural networks without real image data. Cai *et al.* [5] optimized pixel values of generated images to align sample batch statistics with those of batch normalization (BN) layers in full-precision networks. Li *et al.* [27] extended the PTQ framework to transformer architectures by diversifying self-attention across different patches using patch similarity metrics. Meanwhile, deploying PTQ to large language models (LLMs) dynamically search the optimal rounding function for each input sample instead of applying a learnable one, because the activation distributes very significantly across different samples in large models. LLM.int8() [8], SmoothQuant [46] and ZeroQuant [48] handled activation outliers to achieve accurate quantization function learning by eliminating the extreme values with equivalent transformation, yet encountered challenges in effectively scaling to extremely large models due to the unbearable computational costs. Furthermore, GPTQ [12], AWQ [29], and QLoRA [9] deployed low-precision quantization on weight quantization to further reduce the computational complexity. However, these methods employ layer-wise searching strategy to search the rounding functions sequentially, which deviates from the optimal ones due to the lack of cross-block dependency.

## 3  Approach

In this section, we first introduce the preliminaries of post-training quantization for LVLMs and then detail the cross-layer dependency mining for LVLM quantization. Finally, we demonstrate the visual encoder optimization to minimize the quantization errors with negligible search cost overhead.

### 3.1  Post-training Quantization for LVLMs

Network quantization decreases the bitwidth of weights and activations to save computation memory and accelerate inference speed. Conventional quantization-aware training (QAT) optimize all parame-

ters in original full-precision networks for optimal quantization, which is unpractical because of the unacceptable training cost and inaccessibility of the large training datasets. Post-training quantization (PTQ) leverages a small calibration set $X$ to search the optimal threshold in minor rounding functions with frozen network parameters, which significantly reduces data requirements and optimization costs. Specifically, the optimal solution for quantization function learning is acquired by minimizing the distribution discrepancy between quantized outputs and full-precision ones of the entire model. The optimization objective $J$ can be formulated as follows:

$$\min_{\{Q_k\}} \quad J = \left\| W_q^{(n)} X_q^{(n)} - W_r^{(n)} X_r^{(n)} \right\|_2^2$$
$$s.t. \quad X_q^{(k+1)} = Q_k(W_q^{(k)} X_q^{(k)}) \tag{1}$$

where $W_q^{(k)}$ and $X_q^{(k)}$ mean the quantized weights and activations for the $k_{th}$ layer, and $W_r^{(k)}$ and $X_r^{(k)}$ represent their full-precision counterparts. $Q_k$ means the rounding function for the $k_{th}$ layer and $n$ is the total number of the layers in the LVLM. Directly searching the optimal rounding function is NP-hard because the search space increases exponentially with the layer number. Therefore, conventional PTQ methods sequentially search the rounding functions by minimizing the quantization errors for each layer in the greedy way:

$$\min_{Q_k} \quad J = \left\| W_q^{(k)} X_q^{(k)} - W_r^{(k)} X_r^{(k)} \right\|_2^2 \tag{2}$$

where the layer index gradually increases to search the rounding function from bottom to top layers. However, the greedy search ignores the cross-layer dependency of discretization errors, which leads to accumulated discretization errors of model output even for the rounding function with small errors in the bottom layers.

## 3.2 Mining Cross-layer Dependency for LVLM Quantization

Directly search the solution to (1) causes unacceptable search cost, while sequentially search the layer-wise quantization functions results in suboptimal solution. Therefore, we partition the entire model into different blocks consisting of multiple layers. Searching the optimal rounding function by considering the output quantization errors for each block achieves better trade-off between the search cost and the quantization accuracy, which is formulated as follows:

$$\min_{\{Q_k\} \in B_i} \quad J = \left\| W_q^{(L_i)} X_q^{(L_i)} - W_r^{(L_i)} X_r^{(L_i)} \right\|_2^2$$
$$s.t. \quad X_q^{(k+1)} = Q_k(W_q^{(k)} X_q^{(k)}) \tag{3}$$

where $B_i$ represents the $i_{th}$ block in our partition and $L_i$ is the index of the last layer in block $B_i$. Our goal is to obtain the optimal block partition for rounding function search, where we expect the dependency for layers in each block to be strong. Therefore, we can search the rounding functions for layers in each block by minimizing the discretization errors for the block output, and the output errors of the entire model are still minimized.

Explicitly evaluating the cross-layer dependency requires multiple forward pass of LVLMs for given input, where the correlation for discretization errors is calculated from data statistics. To avoid extremely high cost, we have to explore an efficient proxy to approximate the cross-layer dependency. We leverage Information Entropy of the activation to judge the sensitive layers with homogeneous distribution[43]. For sensitive layers, the noise in the former layer caused by the deviation from the global optimal value usually leads to higher deviation for the

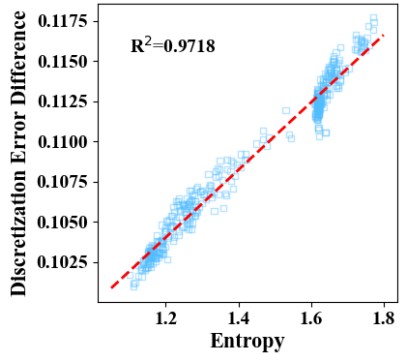

Figure 2: The correlation between discretization error difference (DED) and the activation entropy in 15th layer.

current layer, so that jointly search these layers decrease the block output quantization errors. As a result, discretization error difference (DED) between layer-wise search and joint search is obvious

for sensitive layers with high entropy. In order to reduce the deviation from the optimal rounding points, we should search the quantization function by joint consideration of the current layer and the former one with strong dependency.

Meanwhile, we also empirically verified our assumption shown in Figure 2. Figure 2 is produced with the activations in the 15th layer of the LLaVA architectures on SQA dataset. The horizontal axis demonstrates the entropy of the activations, and the vertical axis depicts the discretization error difference (DED) between layer-wise search and joint search. Different markers mean different input multimodal samples, where DED and the activation entropy are strongly correlated. Therefore, the cross-layer dependency $D(k, k+1)$ between the current layer $k$ and the following layer $k+1$ can be defined as follows:

$$D(k, k+1) = -\sum_{ij} p(x_{q,ij}^{(k)}, x_{q,ij}^{(k+1)}) log p(x_{q,ij}^{(k+1)} | x_{q,ij}^{(k)}) \tag{4}$$

where $\chi^k$ represents the set of possible values about the quantized activation $x_q^k$ and $x_q^{k+1}$ in the $k_{th}$ and $k+1_{th}$ layer, and $p$ means the probability of the variable. $Q_k$ is searched with different quantization levels for rounding function optimization. We leverage uniform quantization to round the full-precision tensors due to the high compatibility with real hardware. In realistic implementation, each element is quantized to the nearest rounding point deterministically. In order to acquire the entropy of the quantized tensor, we approximate the deterministic quantization process with the following distribution:

$$p(x_{q,ij}^{(k)}) = \frac{\exp(-(x_{r,ij}^{(k)} - q_m)^2/\Delta)}{\sum_{m=1}^{M} \exp(-(x_{r,ij}^{(k)} - q_m)^2/\Delta)} \cdot \delta(x_{q,ij}^{(k)} - q_m) \tag{5}$$

where $\Delta$ is the interval between two consecutive rounding points, and $\delta$ represents the pulse distribution. $q_m$ means the $m_{th}$ rounding point in the quantization out of $M$ quantization levels. The cross-layer dependency of two consecutive layers can be depicted by the discretization error difference (DED) between the rounding function searched sequentially and jointly, and larger difference indicates the former layer has significant influence on the discretization errors of the following one. Figure 2 demonstrates the positively correlation between DED and the activation entropy across different input samples with high correlation coefficients. Higher conditional entropy indicates that the activation distribution is homogenized with accumulated quantization errors, which represents larger cross-layer dependency for obvious influence with the following layers. To evaluate cross-layer dependency $D(k_r, k_s)$ of non-consecutive layer $k_r$ and layer $k_s$, we consider the summation of entropy in all intermediate layers between them:

$$D(k_r, k_s) = -\sum_{k=k_r}^{k_s} \sum_{ij} p(x_{q,ij}^{(k)}, x_{q,ij}^{(k+1)}) log(x_{q,ij}^{(k+1)} | x_{q,ij}^{(k)}) \tag{6}$$

Since searching rounding functions within blocks with large number of layers, we constrain the maximum layer numbers for each block. We partition blocks in the LVLM based on the acquired cross-layer dependency:

$$B_i = \{ \bigcup_{k=k_r}^{k_s} Q_k | D(k_r, k_s) > (k_s - k_r)h_0 \} \tag{7}$$

If the average cross-layer dependency between two layers is larger than the threshold $h_0$, all intermediate layers are assigned into a single block for joint rounding function search because the discretization errors of the block output are sensitive to all former layers within a block. Finally, we search the optimal rounding function by minimizing the output discretization errors of each block, where we select the optimal percentile $p$ [6] of the full-precision tensor distribution as the bounds for the uniform quantization.

### 3.3 Optimizing Visual Encoders for LVLM Quantization

LVLMs leverage a visual encoder to extract informative representations for image input, and align visual embedding and text embedding with a projection layer. As the visual encoder significantly

modifies the distribution of the activations in LVLMs, the rounding function in visual encoder can be optimized to minimize the activation entropy to enhance the searching efficiency. As a result, there are fewer layers in each block for rounding function search and the search cost remains low.

Simultaneously minimizing the activation entropy and weakening the cross-layer dependency for all layers causes optimization difficulties, because the large number of layers in LLaMA usually provide conflicted supervision for the visual encoder. Since different layers usually have various influence on the quantization errors of the output from the entire model, we assign different importance weights for the entropy minimization objective across layers which are acquired from the Jacobian:

$$L_{ent} = \sum_{k=1}^{n} ||\frac{\partial E^{(n)}}{\partial X_r^{(k)}}|| \cdot \sum_{ij} p(x_{q,ij}^{(k)}, x_{q,ij}^{(k+1)}) log p(x_{q,ij}^{(k+1)}|x_{q,ij}^{(k)}) \tag{8}$$

where $E^{(n)}$ represents the quantization errors of the final layer in the model. The Jacobian indicates the influence of the current layer to the quantization errors of the final output [34]. Larger Jacobian magnitudes represent the higher influence on the overall discretization errors, and assigning larger weights to those layers can reduce the cross-layer dependency with fast model convergence. Meanwhile, our objective also includes the minimization of discretization errors for both the output of the visual encoder and the LVLM, which can enhance the quantization accuracy for visual representation learning and multi-model reasoning:

$$L_{err} = ||X_q^v - X_r^v|| + \eta ||X_q^{(n)} - X_r^{(n)}|| \tag{9}$$

where $X_q^v$ and $X_r^v$ respectively represent the quantized and full-precision output of the visual encoder, and $\eta$ is a hyperparameter to balance the importance between the discretization errors of the visual encoder and the LVLM. Finally, the overall objective for visual encoder optimization can be written as follows with the hyperparameters $\lambda_1$ and $\lambda_2$:

$$L = L_{reg} + \lambda_1 L_{ent} + \lambda_2 L_{err} \tag{10}$$

where $L_{reg}$ means the auto-regressive loss adopted in training original LVLM to minimize the discrepancy of predicted and target tokens. By optimizing the the visual encoder, we can search the rounding function in more fine-grained blocks with fewer layers to reduce the search cost, while the quantization accuracy still remains high due to the weak cross-layer dependency.

## 4 Experiments

In this section, we conduct extensive experiments for LLaVA and MoE-LLaVA benchmarks on ScienceQA multi-modal question answering dataset to evaluate the effectiveness of our methods. We first introduce the implementation details of our method. We then conduct ablation studies to evaluate the effectiveness of cross-layer dependency mining and visual encoder optimization. Finally, we compare our Q-VLM with the state-of-the-art post-training quantization methods to show its superiority.

### 4.1 Implementation Details

We utilize the large vision-language frameworks for post-training quantization including LLaVA [31] and MoE-LLaVA [28] with their pre-trained weights for multi-modal question answering tasks. We set the bitwidth of quantized weight and activation to 6 and 4 to evaluate our method in different quality-efficiency trade-offs uniform quantization scheme where the interval between adjacent rounding points was equal. We followed the initialization of the quantization function parameters in QLoRA [9] for the baseline methods and our Q-VLM, where we minimized the lp distance [37, 29] between the full-precision and quantized activations to optimize the value range for clipping. We set the maximum layer depth to 3 within a block to achieve satisfying trade-offs between the discretization errors and the search cost. In the LVLM quantization exploration, we adjust hyperparameters $p$ of percentile ranging from 1.0 to 0.98 with 0.005 interval for cross-layer dependency mining. Meanwhile, we modified the hyperparameter $\eta$ to demonstrate the effect of the discretization loss which optimizes the visual encoder in 9. For the parameter learning in LVLM quantization, we randomly select 64 vision-language pairs from the datasets for hyper-network learning where the batchsize was assigned with 8 for calibration set construction. The quantization function parameters were updated for 10

| Method | W6A6 | | | W4A4 | | |
|---|---|---|---|---|---|---|
| | Memory | Search cost | Accuracy | Memory | Search cost | Accuracy |
| QLoRA | | 23.7 | 88.43 | | 23.5 | 77.53 |
| +CDM | 9.7G | 26.9 | 88.95 | 6.6G | 25.9 | 78.66 |
| +VEO | | 24.1 | 88.72 | | 23.7 | 78.35 |
| Q-VLM | | 25.1 | 89.34 | | 24.6 | 79.79 |

Table 1: Effect of different LVLM quantization method we proposed. "CDM" means cross-layer dependency mining and "VEO" stands for visual encoder optimization. We report the result of LLaVA-7B model on ScienceQA dataset.

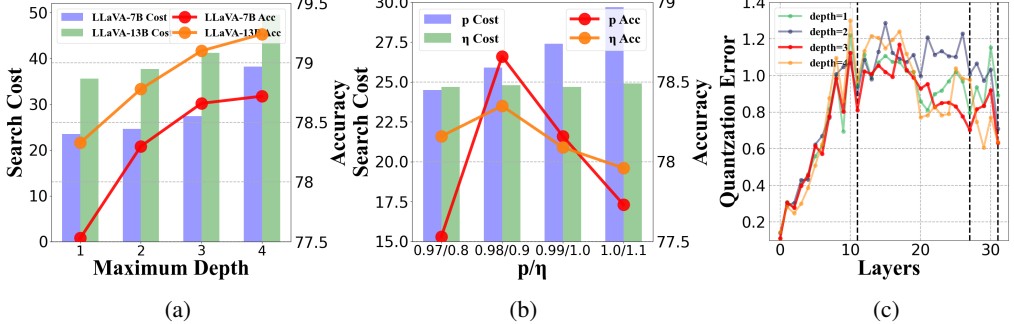

(a)          (b)          (c)

Figure 3: (a)The answering accuracy and searching cost w.r.t. different maximum layer depth within a block. (b) The answering accuracy and searching cost w.r.t. different hyperparameters across various vision-language models. (c) Quantization errors w.r.t. different maximum layer depth across various layers.

epochs in searching process, and the acquired discretization function was directly employed for multi-modal question answering. The multi-modal answer reasoning dataset is ScienceQA [35], which contains 21k vision-language multiple choice questions. We also contain VizWiz [15] and VQA-v2 [14] datasets.

## 4.2 Ablation Study

Since previous layer-wise searching methods ignore cross-layer dependency, we employ joint searching strategy with dependency mining. In order to investigate the influence of the block-wise searching strategy, we vary the maximum number of layer depths contained in a single block with different trade-off between discretization errors and searching cost. Meanwhile, we adjust the searching space of percentile by modifying the hyperparameter $p$ and $\eta$ for cross-layer dependency mining and visual encoder optimizing to select the optimal clipping range in quantization function. Finally, we randomly select 64 vision-language pairs from ScienceQA dataset to finetune quantized models and demonstrate the effectiveness of the proposed cross-layer dependency mining and visual encoder optimizing method. All experiments in the ablation study were conducted with ScienceQA dataset and the LLaVA-v1.3-7B framework.

**Performance w.r.t. the maximum layer depth within a block:** Integrating multiple layers with cross-layer dependency into a single block can exert considerable influence on the clipping and rounding errors throughout the entire model, albeit at the expense of exponentially escalating search costs. Figure 5a illustrates the answering accuracy for our method that astrict different maximum joint layers, where the performance enhancement for layer depth exceeds 3 is slight with significantly increased complexity overhead. To ensure efficient quantization of LVLMs with sizable accuracy increase, we assign the maximum layer depth within a block to 3 in subsequent experiments.

**Performance w.r.t. different method we proposed in question answering process:** To verify the effectiveness of different method we proposed in LVLM quantization, we conduct the ablation study on ScienceQA dataset under different bitwidth. Table 4 illustrates the memory usage in inference, searching cost in calibration, and answering accuracy for our method under different bitwidth for LLaVA-v1.3-7B model. Observing the second rows, the cross-layer dependency mining(CDM) module is important for the final performance, because mining the cross-layer dependency and

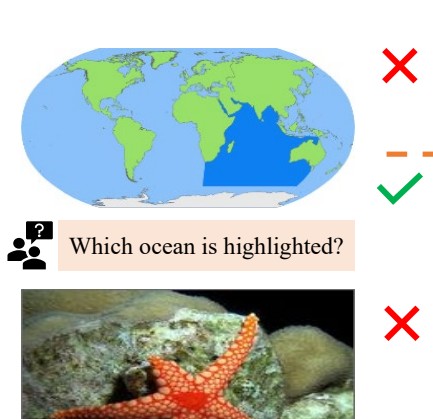

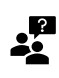 Which ocean is highlighted?

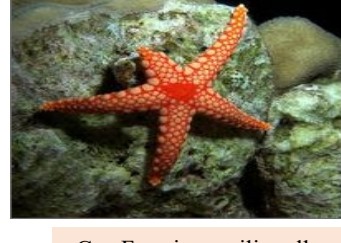

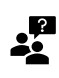 Can Fromia monilis cells make their own food?
Options: (A) yes (B) no

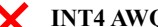 **INT4 AWQ**
The Southern Ocean reaches from the shores of Antarctica to South latitude. **The answer is B.**

- - - - - - - - - - - - - - - - - - - - - - -

**INT4 Q-VLM**
Oceans are huge bodies of salt water. The world has five oceans. **This is the Indian Ocean. The answer is A.**

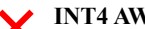 **INT4 AWQ**
Animals get their food by digesting other organisms. **But in the 1950s, scientists discovered that animals can make their own food.** Fromia monilis cells use chemosynthesis to make their food. **The answer is A.**

- - - - - - - - - - - - - - - - - - - - - - -

**INT4 Q-VLM**
Today, many scientists classify organisms into six broad kingdoms. The table below shows some traits used to describe each kingdom. **Fromia monilis is an animal.** Animal cells cannot make their own food. **The answer is B.**

|  | **Bacteria** | **Archaea** | **Protists** | **Fungi** | **Animals** | **Plants** |
|---|---|---|---|---|---|---|
| How many cells do they have? | one | one | one or many | one or many | many | many |
| Do their cells have a nucleus? | no | no | yes | yes | yes | yes |
| Can their cells make food? | some species can | some species can | some species can | no | no | yes |

Figure 4: Visual reasoning examples from LLaVA-13B model. Q-VLM improves over the AWQ baseline for W4A4 quantization, reducing quantization errors and providing more reasonable answers. We color the text to show the correct or wrong responses.

leveraging block-wise optimization strengthen the cooperation between layers and minimize the overall quantization errors. Visual encoder optimization (VEO) without cross-layer dependency also significantly modifies the distribution of the activations in LVLMs for discretization function learning. Q-VLM disentangles the cross-layer dependency for fine-grained search space decomposition, so that precise rounding functions can be acquired with further reduced search cost.

**Performance w.r.t. hyperparameters $p$ and $\eta$:** The hyperparameters $p$ control the search space for cross-layer dependency mining and $\eta$ balances the importance between the discretization errors of the visual encoder and the LVLM. Larger $p$ leads to reduced rounding errors by diminishing the influence of outliers, although excessively large values result in excessive clipping of significant information in data distribution. Figure 5b depicts the answering accuracy for different hyperparameter settings, where the medium value for both parameters achieves the highest performance and achieves the trade-off between quantization errors and searching space.

**Visualization reasoning examples:** We further provide qualitative visual reasoning example of the LLaVA-v1.3-13B model in Figure 4 from ScienceQA dataset. Q-VLM improves the responses compared to AWQ baseline for W4A4 quantization setting, leading to more reasonable and particular answers for vision-language question pairs. In this first example, Q-VLM correctly answers the question about the highlighted ocean, while AWQ appears to have limited comprehension of the image information. For the second example, AWQ under W4A4 lost substantial information to produce sound reasoning about whether Fromia monilis cells make their own food, while Q-VLM answered correctly and even produced a table for precise classification about the six broad organisms called kingdom. Q-VLM improves the visual reasoning ability of LVLMs by reducing factual errors in the responses.

| | Bits | Method | Subject | | | Context Modality | | | Average |
|---|---|---|---|---|---|---|---|---|---|
| | | | NAT | SOC | LAN | TXT | IMG | NO | |
| LLaVA-7B | FP | - | 89.39 | 96.06 | 85.64 | 88.71 | 87.65 | 88.50 | 89.81 |
| | W6A6 | AWQ | 85.39 | 92.01 | 83.27 | 84.80 | 83.54 | 85.99 | 86.23 |
| | | QLoRA | 88.45 | 94.71 | 84.45 | 87.63 | 86.07 | 87.87 | 88.43 |
| | | Q-VLM | 89.43 | 95.73 | 84.00 | 88.71 | 87.51 | 87.25 | **89.34** |
| | W4A4 | AWQ | 74.33 | 72.22 | 74.82 | 73.41 | 67.13 | 77.98 | 74.02 |
| | | QLoRA | 77.53 | 75.48 | 79.18 | 76.64 | 70.70 | 81.95 | 77.53 |
| | | Q-VLM | 80.86 | 75.93 | 80.73 | 80.01 | 72.48 | 83.90 | **79.79** |
| LLaVA-13B | FP | - | 90.19 | 93.14 | 87.09 | 89.39 | 87.06 | 89.83 | 90.00 |
| | W6A6 | AWQ | 88.03 | 92.60 | 84.00 | 86.02 | 85.18 | 86.41 | 87.57 |
| | | QLoRA | 88.87 | 92.89 | 85.64 | 87.59 | 86.56 | 87.53 | 88.87 |
| | | Q-VLM | 89.54 | 93.18 | 86.50 | 88.12 | 87.01 | 88.85 | **89.70** |
| | W4A4 | AWQ | 80.71 | 70.61 | 78.49 | 79.46 | 70.76 | 81.82 | 77.91 |
| | | QLoRA | 79.62 | 71.43 | 82.45 | 78.25 | 68.42 | 85.30 | 78.64 |
| | | Q-VLM | 82.55 | 73.32 | 83.18 | 81.03 | 70.82 | 86.74 | **80.78** |
| MoE-LLaVA-1.6B | FP | - | 64.01 | 58.57 | 63.30 | 62.80 | 54.78 | 66.97 | 62.68 |
| | W6A6 | AWQ | 59.83 | 57.78 | 61.48 | 58.94 | 52.95 | 64.25 | 59.83 |
| | | QLoRA | 62.98 | 57.78 | 61.85 | 61.87 | 53.99 | 65.37 | 61.60 |
| | | Q-VLM | 64.14 | 58.68 | 62.85 | 62.80 | 55.23 | 66.69 | **62.46** |
| | W4A4 | AWQ | 53.69 | 49.58 | 54.27 | 52.52 | 47.81 | 57.65 | 52.98 |
| | | QLoRA | 54.48 | 49.69 | 55.55 | 53.20 | 47.30 | 57.58 | 53.24 |
| | | Q-VLM | 55.06 | 51.94 | 56.27 | 54.57 | 48.03 | 58.34 | **54.72** |

Table 2: Comparisons with the state-of-the-arts post-training quantization methods for LLaVA-v1.3 and MoE-LLaVA models across bitwidth setting.Results (accuracy) on Science QA dataset. Question classes: NAT = natural science, SOC = social science, LAN = language science, TXT = text context, IMG = image context, NO = no context.

## 4.3  Comparison with the State-of-the-art Methods

In this section, we compare our proposed method with the state-of-the-art post-training quantization frameworks. As far as we know, we are the first to complete multi-modal LVLMs under W4A4 setting, so we conduct a series of baseline methods by combining the conventional state-of-the-art post-training quantization methods for fair comparison. For weight quantization, we follow the experiment setting of AWQ [29] and QLoRA [9]. Meanwhile, as the activations in language modal exhibit significant variations in value range across different channels, we reproduce RPTQ [50] with per-channel activation quantization. As the activation distribution appears larger variance across different tokens in vision modal, we utilize per-token quantization following Outlier Suppression [44]. The answering accuracy of the baseline methods is acquired by implementing the officially released code and pre-trained model.

Table 2 shows the comparison of top-1 accuracy of different post-training quantization methods across various LVLMs architectures including LLaVA-v1.3-7B, LLaVA-v1.3-13B [31] and MoE-LLaVA-1.6B [28], where bitwidths of weights for quantized layers select from 4 and 6. AWQ searched the optimal scale to protect the salient weight channels and decreased the quantization errors for weight quantization, while weight-only quantization scaling significantly increased outlier in activation and led to significant quantization loss. QLoRA with RPTQ employs sequentially search the layer-wise rounding functions by minimizing activation discretization errors. However, ignoring cross-layer dependency of discretization errors fails to acquire the optimal rounding strategy and degrades the performance significantly. On the contrary, our Q-VLM mines the cross-layer dependency of output distribution across layers and decomposes the large search space from the entire model to blocks containing multiple layers, which minimizes the block-wise discretization errors to avoid suboptimal quantization, and further optimizes the visual encoder to disentangle the cross-layer dependency for fine-grained search space decomposition. As a result, our method outperforms QLoRA by 2.26 (79.79 vs. 77.53) for answering accuracy in ScienceQA dataset under 4-bit in LLaVA-7B model. The computational cost remains the same for baseline methods and our Q-VLM due to the stored rounding parameters. The advantage of our method becomes more obvious for 4-bit LVLMs because quantization errors and cross-layer dependency are more important for networks with low capacity.

| Model | Dataset | FP | W6A6 | | | W4A4 | | |
|---|---|---|---|---|---|---|---|---|
| | | | AWQ | QLoRA | Q-VLM | AWQ | QLoRA | Q-VLM |
| LLaVA-7B | SQA | 66.79 | 65.87 | 66.16 | **66.67** | 56.73 | 56.50 | **57.70** |
| | VizWiz | 49.87 | 48.51 | 49.23 | **49.86** | 44.29 | 44.73 | **45.82** |
| | VQA v2 | 78.50 | 77.51 | 77.58 | **78.52** | 71.89 | 72.02 | **72.51** |
| LLaVA-13B | SQA | 71.61 | 71.37 | 71.52 | **72.27** | 68.13 | 68.04 | **68.84** |
| | VizWiz | 53.63 | 52.35 | 52.85 | **53.69** | 47.55 | 47.97 | **49.28** |
| | VQA v2 | 79.94 | 78.83 | 79.25 | **79.65** | 71.55 | 72.19 | **73.02** |

Table 3: Comparisons with the state-of-the-arts post-training quantization methods for LLaVA-v1.5 models in various VQA datasets across bitwidth setting.

| Method | FP | | | W8A8 | | | W4A4 | | |
|---|---|---|---|---|---|---|---|---|---|
| | Time | Memory | Accuracy | Time | Memory | Accuracy | Time | Memory | Accuracy |
| QLoRA | | | | 16.7h | 16.5G | 89.32 | 17.0h | 10.7G | 78.64 |
| AWQ | 12.9h | 24.0G | 90.00 | 11.2h | 17.2G | 88.94 | 8.9h | 11.2G | 77.91 |
| Q-VLM | | | | **11.2h** | **15.7G** | **89.82** | **8.9h** | **9.6G** | **80.78** |

Table 4: Comparisons with the state-of-the-arts post-training quantization methods for LLaVA-v1.3-13B models about inference time, memory and accuracy in Science QA dataset.

We also evaluate our method on other datasets with different architectures to verify the generalization ability. Table 3 demonstrates the accuracy of different post-training quantization methods of LVLMs on the VQA dataset. Our method achieves the highest accuracy on different datasets, which means that our method can be robustly deployed in diverse downstream tasks. Table 4 depicts the efficiency and accuracy of different methods. Both AWQ and our method can reduce the search time and the inference memory compared with the original full-precision LVLMs, while our method can further reduce the memory because of the additional quantization of the CLIP. When deploying the quantized LVLMs on mobile devices, our method is more practical due to the limited memory footprint.

## 5    Conclusion

In this paper, we have presented a novel post-training quantization framework of large vision-language models for efficient multi-modal reasoning. Different from conventional methods which sequentially search the layer-wise rounding functions without considering cooperation between layers, we mine the cross-layer dependency and decompose the large search space from the entire model to blocks containing multiple layers. The proxy of entropy prompts the efficient and optimal block partition for rounding function search with the goal of minimizing the block-wise discretization errors. Therefore, the quantized model remains competitive performance with original full-precision counterparts while the search cost is low. Moreover, we optimize the visual encoder to disentangle the cross-layer dependency for fine-grained search space decomposition, so that precise rounding functions can be acquired with further reduced search cost. Extensive experiments demonstrate that our methods highly compress the large vision-language models without performance degradation and achieve higher answer reasoning ability than the state-of-the-art post-training quantization methods across large vision-language models with various architectures even under W4A4.

**Limitations:** One limitation of our work is that applying our framework to the setting of extremely low bitwidths degrade the performance very significantly, which is also a common issue in post-training quantization of foundation models. As foundation models are much larger than the resource limit of mobile devices, we will design efficient post-training quantization method to deploy the LVLMs on embedded equipment such as cellphones and wearable devices.

## Acknowledgement

This work was supported in part by the National Key Research and Development Program of China under Grant 2023YFF1105101 and in part by the Beijing Natural Science Foundation under Grant No. L247009.

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

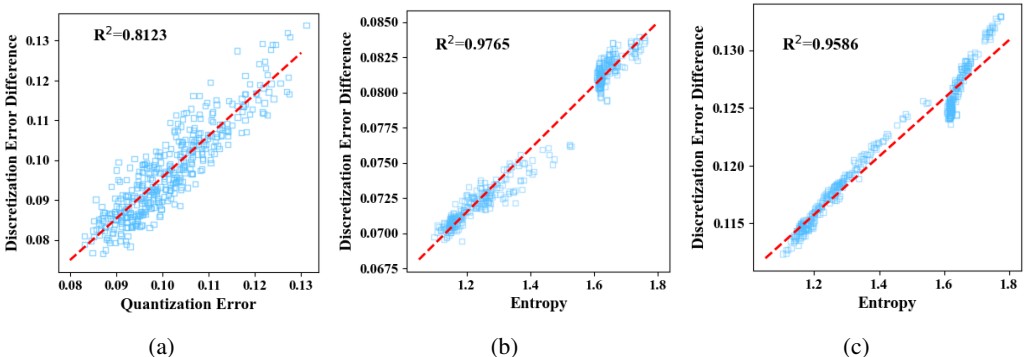

Figure 5: (a) The correlation between discretization error difference (DED) and the quantization errors in 15th layer. (b) The correlation between DED and the entropy in 5th layer and (c) in 25th layer.

| Model | Dependency Proxy | W6A6 | | W4A4 | |
|---|---|---|---|---|---|
| | | Accuracy | Search Cost | Accuracy | Search Cost |
| LLaVA-7B | Quantization Errors | 88.59 | 43.7 | 77.97 | 41.2 |
| | Entropy | 88.95 | 26.9 | 78.66 | 25.9 |
| LLaVA-13B | Quantization Errors | 88.96 | 58.6 | 78.82 | 54.2 |
| | Entropy | 89.04 | 33.1 | 79.89 | 31.8 |

Table 5: Comparisons with different proxy for mining cross-layer dependency for LLaVA-v1.3 models in ScienceQA dataset across bitwidth setting.

## A    Why leveraging entropy as proxy:

The benefit and motivation for using entropy rather than quantization errors as a proxy for block-wise searches lie in several key considerations. We analyze that larger entropy indicates more homogeneous data distribution, which is a well-established principle in information theory. Consequently, DED and activation entropy are strongly correlated with an value of 0.97. However, greater quantization error does not necessarily imply more homogeneous data distribution and does not show a positive correlation with DED, having an value of 0.81, which is empirically verified in the figure 5.

Meanwhile, the search cost of quantization errors doubles compared with entropy as a proxy, as the calculation of quantization errors requires multiple forward passes for both the FP model and the quantized model. The weak correlation and the unbearable search cost render quantization error unsuitable as a metric for measuring cross-layer dependency.

Furthermore, we conducted experiments comparing the proxy effectiveness of quantization error and entropy across different models under various bitwidths in Table 5. Entropy outperformed quantization errors by a significant margin (78.66 vs. 77.97), showing a strong cross-layer dependency within each block. This allowed us to achieve optimal block partitioning by mining the cross-layer dependency.

## B    Performance on more baseline methods

We have extended our experiments to an additional baseline method ZeroQuant-V2[47] and compared it against our proposed methods in Table 6. ZeroQuant-V2 leverages per-token quantization with different rounding functions to minimizing activation discretization errors. However, ignoring cross-layer dependency of discretization errors fails to acquire the optimal rounding strategy with severe outliers under low bitwidth and degrades the performance significantly. On the contrary, our Q-VLM mines the cross-layer dependency of output distribution across layers, minimizing the block-wise discretization errors to avoid suboptimal quantization. We further optimize the visual encoder to disentangle the cross-layer dependency for fine-grained search space decomposition. As a

| Model | Quantization Method | W8A8 | | W4A4 | |
|---|---|---|---|---|---|
| | | Accuracy | Inference Time | Accuracy | Inference Time |
| LLaVA-7B | ZeroQuant-V2 | 89.04 | 10.7h | 78.08 | 7.3h |
| | Q-VLM | 89.58 | 8.3h | 79.79 | 6.1h |
| LLaVA-13B | ZeroQuant-V2 | 89.13 | 12.6h | 78.81 | 9.7h |
| | Q-VLM | 89.81 | 11.2h | 80.78 | 8.9h |

Table 6: Comparisons with different quantization methods for 7B and 13B models across W6A6 and W4A4 bitwidth settings.

| Dataset | Shots | FP | 8bit | | 4bit | |
|---|---|---|---|---|---|---|
| | | | Q-LoRA | Q-VLM | Q-LoRA | Q-VLM |
| Vizwiz | 0 | 23.79 | 21.24 | 21.47 | 17.62 | 18.69 |
| | 4 | 27.05 | 25.83 | 26.59 | 24.17 | 24.55 |
| | 32 | 39.76 | 36.38 | 37.60 | 31.64 | 35.52 |
| Hateful Memes | 0 | 50.23 | 47.75 | 49.12 | 43.86 | 44.22 |
| | 4 | 50.10 | 48.62 | 49.55 | 45.12 | 45.26 |
| | 32 | 50.27 | 50.02 | 51.05 | 45.76 | 47.84 |

Table 7: Performance comparison on Vizwiz and Hateful Memes datasets across FP, 8bit, and 4bit quantization methods with different shot settings.

result, our method outperforms ZeroQuant-V2 by 1.71 (79.79 vs. 78.08) in answering accuracy on ScienceQA dataset under 4-bit in LLaVA-7B model. Additionally, our method enhances inference speed, exceeding ZeroQuant-V2 by 1.2h (6.1h vs. 7.3h) due to utilizing stored rounding parameters instead of dynamic per-token quantization. The additional baseline provides a more comprehensive evaluation framework to highlight the strengths of our approach.

## C  Performance on other multi-modal architectures

We also explored the multi-modal architecture OpenFlamingo[3] to ensure the robustness and generalizability of our methods 7. We deploy our method on OpenFlamingo 3B model using Vizwiz and Hateful Memes[22] datasets, selecting bitwidths of 4 and 8 for quantized layers. Q-VLM designed in LLaVA-like architectures can be effectively adapted to cross-attention based VLMs due to the consistent core mechanism of cross-attention and the robust multimodal alignment capabilities pre-trained on large-scale vision-language pairs. Since OpenFlamingo is a cross-attention based VLM, exploiting cross-layer dependency is particularly suitable. Our method outperforms Q-LoRA by 1.22 (37.60 vs. 36.38) under 8-bit in OpenFlamingo-3B model. The advantage of our method becomes more obvious for 4-bit 3B LVLMs because quantization errors and cross-layer dependency play a more significant role in networks with low capacity. These results underscore the robustness and generalizability of our approach across different tasks, model architectures and datasets, demonstrating its effectiveness in diverse scenarios.

