# OpenReview forum: "Q-VLM: Post-training Quantization for Large Vision-Language Models"
_NeurIPS.cc/2024/Conference — NeurIPS 2024 poster_

### Official Review · Reviewer_rcqj · 2024-07-11

**Soundness:** 3
**Presentation:** 4
**Contribution:** 3
**Rating:** 5
**Confidence:** 2

**Summary:**

The authors present a novel post-training quantization framework for large multimodal language models to enhance inference speed. This method accounts for cross-layer dependencies that significantly impact overall model discretization errors and leverages activation entropy to effectively balance these errors with search costs. Extensive experimental results demonstrate that this framework can reduce memory usage and improve generation speed by 1.44x on the 13B LLaVA model, while maintaining comparable performance across diverse multimodal downstream tasks.

**Strengths:**

1. The paper considers cross-layer dependencies during the quantization process for the first time, which is novel.

2. The proposed quantization method is simple and effective. It significantly reduces memory usage and improves inference speed, which is crucial for MLLM deployment.

3. The paper is well-written and easy to follow.

**Weaknesses:**

1. Is there any quantization for the adapter between the vision encoder and the LLM? With various types of adapters such as MLP for LLaVA, Q-former for BLIP-2 [1], and Visual Abstractor for mPLUG-Owl [2], does the proposed method apply to models with different adapters?

2. Instead of comparing with outdated methods such as Q-LoRA and AWQ, it would be beneficial to compare with more advanced techniques like ZeroQuant-FP [3] and include a more in-depth discussion.

[1] Li, Junnan, et al. "Blip-2: Bootstrapping language-image pre-training with frozen image encoders and large language models." International conference on machine learning. PMLR, 2023.

[2] Ye, Qinghao, et al. "mplug-owl: Modularization empowers large language models with multimodality." arXiv preprint arXiv:2304.14178 (2023).

[3] Wu, Xiaoxia, Zhewei Yao, and Yuxiong He. "Zeroquant-fp: A leap forward in llms post-training w4a8 quantization using floating-point formats." arXiv preprint arXiv:2307.09782 (2023).

**Questions:**

1. Does this quantization method only work for LLaVA-like architectures? How does it apply to cross-attention based VLMs such as Flamingo [1] and Otter [2]?

[1] Alayrac, Jean-Baptiste, et al. "Flamingo: a visual language model for few-shot learning." Advances in neural information processing systems 35 (2022): 23716-23736.

[2] Li, Bo, et al. "Mimic-it: Multi-modal in-context instruction tuning." arXiv preprint arXiv:2306.05425 (2023).

**Limitations:**

The authors have addressed the limitation in the manuscript.

---

> ### Author Rebuttal · Authors · 2024-08-06
>
> We appreciate your suggestion and agree that comparisons with more advanced techniques and cross-attention based VLM architectures would provide a deeper insight into the effectiveness of our method. Below are our detailed responses.
>
> **Q1: Ablation study about the projectors.**
>
> **[Reply]** For quantization about the projector between the vision encoder and the LLM, we did not quantize it as it contains only 20M parameters which is negligible compared to the 7B parameters of the entire model. Additionally, remaining the projector to FP would not result in significant increase in inference speed by **0.06h (6.13h vs. 6.07h)** compared with 4-bit projector. We conduct experiments on OpenFlamingo architecture with encoder-decoder structure which emphasize the leverage of the model inherent architecture for multimodal feature alignment and our Q-VLM still achieves outstanding performance compared with baseline methods. The results are presented as follows based on Tables in Overall Author Rebuttal Q1:
> |Model|Dataset|FP|8bit ZeroQuant-V2|8bit Q-VLM|4bit ZeroQuant-V2|4bit Q-VLM|
> |---|---|---|---|---|---|---|
> |LLaVA-7B|ScienceQA|89.81|89.04|89.58|78.08|**79.79**|
> |LLaVA-13B|ScienceQA|90.00|89.13|89.81|78.81|**80.78**|
> |**Model**|**Dataset**|**FP**|**8bit Q-LoRA**|**8bit Q-VLM**|**4bit Q-LoRA**|**4bit Q-VLM**|
> |OpenFlamingo-3B|Vizwiz|39.76|36.38|37.60|31.64|**35.52**|
> ||Hateful Memes|50.27|50.02|51.05|45.76|**47.84**|
>
> We conclude that our Q-VLM designed in LLaVA-like architectures can be effectively adapted to other transformer-based VLM architectures with different projectors due to the effectively vision-language alignment information preserved in projectors.
>
> **Q2: Performance on more baseline methods.**
>
> **[Reply]** In response to **Overall Author Rebuttal Q1**, we have conducted experiments and include a more in-depth discussion with more advanced techniques ZeroQuant-V2 compared with proposed Q-VLM. ZeroQuant-V2 fails to acquire the optimal rounding strategy with severe outliers under low bitwidth led to significant quantization loss. On the country, our Q-VLM leverages Information Entropy as a proxy and mines the cross-layer dependency to achieve optimal block partitioning. As a result, our method outperforms ZeroQuant-V2 by **1.71 (79.79 vs. 78.08)** in answering accuracy on ScienceQA dataset under 4-bit in LLaVA-7B model. Additionally, our method enhances inference speed, exceeding ZeroQuant-V2 by 0.06h (6.13h vs. 6.07h) due to utilizing stored rounding parameters instead of dynamic per-token quantization. The additional baseline provides a more comprehensive evaluation framework to highlight the strengths of our approach.
>
> **Q3: Performance on other multi-modal architectures.**
>
> **[Reply]** In response to **Overall Author Rebuttal Q1**, we additionally explored our Q-VLM into another multi-modal architecture OpenFlamingo. Q-VLM designed in LLaVA-like architectures can be effectively adapted to cross-attention based VLMs due to the consistent core mechanism of cross-attention and the robust multimodal alignment capabilities pre-trained on large-scale vision-language pairs. Since OpenFlamingo is a cross-attention based VLM, exploiting cross-layer dependency is particularly suitable. Our method outperforms Q-LoRA by **2.08 (47.84 vs. 45.76)** under 4-bit in OpenFlamingo-3B model. Q-VLM achieves high accuracy on different cross-attention based multi-modal architectures, which means that our method maintain effectiveness and generalizability.

---

> ### Comment · Reviewer_rcqj · 2024-08-12
>
> Hi. Thank you for writing the rebuttal! I confirm I have read the rebuttal, which addresses most of my concern. So I would keep my score.

---

### Official Review · Reviewer_EGvn · 2024-07-17

**Soundness:** 3
**Presentation:** 3
**Contribution:** 2
**Rating:** 6
**Confidence:** 2

**Summary:**

This paper proposes Q-VLM, a post-training quantization framework for Large Vision-Language Models (LVLMs). It aims to reduce the model complexity of LVLMs for practical deployment by replacing float numbers with quantized ones and substituting multiply-accumulate operations with integer arithmetic. The key innovation lies in mining cross-layer dependency to efficiently search for optimal rounding functions that minimize quantization noise across the entire model. The authors also optimize the visual encoder to further reduce search costs and maintain quantization accuracy. Experimental results on LLaVA and MoE-LLaVA models demonstrate significant memory compression and speed increases without severe performance degradation.

**Strengths:**

1. **Novel Approach to Quantization**: The paper introduces a novel approach to post-training quantization that considers cross-layer dependencies, which is a significant departure from traditional layer-wise methods. This approach has the potential to improve the efficiency and accuracy of quantized LVLMs.
2.  **Clear Presentation**: The paper is well-organized and clearly written, making it easy to follow the authors' thought process and understand their contributions.

**Weaknesses:**

1. **Limited Evaluation on Diverse Datasets**: The majority of experiments are conducted on the ScienceQA dataset, which may not fully represent the diverse range of tasks and challenges that LVLMs encounter in real-world applications. Evaluating the method on a wider range of datasets would provide a more comprehensive assessment of its effectiveness and generalizability.
2. **Marginal Performance Improvement**: The observed improvements in some cases are relatively small, potentially due to the specific characteristics of the ScienceQA task. It would be beneficial to investigate whether the method's impact is consistent across different tasks and datasets, or if its effectiveness is limited to specific scenarios.

Some typo:

In the abstract and introduction, the phrase "compresses the memory by 2.78x and increase the generate speed by 1.44x **about** 13B LLaVA model" may be unclear. It could be revised to "compresses the memory by 2.78x and increases the generation speed by 1.44x **for** the 13B LLaVA model" for improved clarity.

**Questions:**

1. Can the authors provide more experiments on other tasks/datasets to demonstrate the generalizability of their method beyond ScienceQA?

**Limitations:**

Not found.

---

> ### Author Rebuttal · Authors · 2024-08-06
>
> Thank you for careful reading and valuable comments. We will check the paper carefully, and modify the presentation of the ambiguous parts in the final version. We provide answers to the questions as follows:
>
> **Q1: Evaluation on Diverse Datasets.**
>
> **[Reply]** LVLMs encounter diverse range of tasks and challenges in real-world applications. We evaluate our proposed method on various datasets with different architectures to verify the effectiveness and generalizability. The experiments are presented in Table 3 and as follows:
> |Model|Dataset|FP|6bit AWQ|6bit Q-LoRA|6bit Q-VLM|4bit AWQ|4bit Q-LoRA|4bit Q-VLM|
> |---|---|---|---|---|---|---|---|---|
> |LLaVA-7B|MM-Vet [1]|31.40|30.79|31.03|**31.59**|28.17|28.42|**29.28**|
> ||ScienceQA|66.79|65.87|66.16|**66.67**|56.73|56.50|**57.70**|
> ||VizWiz|49.87|48.51|49.23|**49.86**|44.29|44.73|**45.82**|
> ||VQA v2|78.50|77.51|77.58|**78.52**|71.89|72.02|**72.51**|
> |LLaVA-13B|MM-Vet [1]|36.07|34.78|34.67|**35.83**|30.16|30.71|**31.64**|
> ||ScienceQA|71.61|71.37|71.52|**72.27**|68.13|68.04|**68.84**|
> ||VizWiz|53.63|52.35|52.85|**53.69**|47.55|47.97|**49.28**|
> ||VQA v2|79.94|78.83|79.25|**79.65**|71.55|72.19|**73.02**|
>
> ScienceQA challenges LVLMs with domain-specific question answering requiring deep understanding of scientific concepts and visual data interpretation. VQA v2 and MM-Vet[1] tasks LVLMs with the need to accurately answer open-ended questions about diverse images, testing their ability to integrate visual and linguistic information. VizWiz leverage LVLMs to interpret and answer questions about everyday images captured by visually impaired users, often dealing with low-quality and diverse real-world content. Our method outperforms Q-LoRA by **1.31 (49.28 vs. 47.97)** under 4-bit in LLaVA-v1.5-13B model on VizWiz dataset, which shows the superiority of mining cross-layer dependency to effectively reduces quantization errors and enhances generalizability for low-quality and diverse real-world content. Q-VLM achieves the highest accuracy among various post-training quantization methods for LVLMs across different VQA datasets, indicates that our method can be robustly deployed in diverse downstream tasks.
>
> **Q2: Evaluation on Different Tasks.**
>
> **[Reply]** Our Q-VLM has achieved the highest accuracy across different visual question answering tasks including ScienceQA, VizWiz, VQA v2 and MM-Vet, where our Q-VLM outperforms Q-LoRA by 0.93 (31.64 vs. 30.71) under 4-bit in LLaVA-v1.5-13B model on MM-Vet dataset.
>
> Additionally, we conduct experiments on other multi-modal architecture OpenFlamingo with Hateful Memes dataset for classification task to further demonstrate effectiveness and generalizability of our proposed method in the **Overall Author Rebuttal Q1**. Our method outperforms Q-LoRA by 2.08 (47.84 vs. 45.76) and 1.03 (51.05 vs. 50.02) in OpenFlamingo-3B model under 4-bit and 6-bit respectively. The Hateful Memes dataset is a collection of multimodal data containing images paired with text captions, specifically designed for research in detecting hate speech in memes. Our Q-VLM excels in the classification task on the Hateful Memes dataset by effectively combining vision-language information to accurately identify hate speech in multimodal memes.
>
> **Q3: Correcting Some Typos.**
>
> **[Reply]** Thank you for your valuable comments. We have thoroughly revised the paper, carefully improving the writing, and correcting misleading wording and grammatical errors. We hope these revisions make the writing more professional and easier to understand.
>
> [1] Yu, Weihao, et al. "Mm-vet: Evaluating large multimodal models for integrated capabilities." arXiv preprint arXiv:2308.02490 (2023).

---

> > ### Comment · Reviewer_EGvn · 2024-08-08
> > **Response to Rebuttal**
> >
> > Prior to the rebuttal, my primary concern was the universal effectiveness of the proposed Q-VLM methodology. I am pleased to see the additional experimental results across various tasks, datasets, and models, all of which support the superior performance of Q-VLM. As such, I am raising my score from 4 to 6.

---

> > > ### Author Response · Authors · 2024-08-08
> > >
> > > Thank you so much for your valuable feedback on our paper. We greatly appreciate your insights and will incorporate your suggestions to enhance the quality of our work.

---

### Official Review · Reviewer_ff1U · 2024-07-19

**Soundness:** 2
**Presentation:** 2
**Contribution:** 3
**Rating:** 5
**Confidence:** 4

**Summary:**

The authors propose a new post-training quantization method for LVLMs. The authors separate several layers in a LVLM into blocks and search for the optimal quantization bitwidth for each block individually. The authors also introduced a new objective function for quantizing the vision encoder. The extensive experimental results show the proposed methods outperform the SOTA post-training quantization methods on LLaVA models.

**Strengths:**

- The authors provide a relatively comprehensive related work discussion in the paper.
- The paper is well-motivated. Post-training quantization on LLM / VLM is critical for deploying these models due to their extensive computation consumption during training.
- The authors provide a large amount of evaluations and ablations in the experimental section to help other researchers better understand their model performance.

**Weaknesses:**

- The overall methods part is difficult to follow. It is not clear if the authors adopt different quantization strategies in visual encoders and language model and the projects in LLaVA. Also, it is not clear to me why the authors design those specific methods for LVLM, i.e., why the proposed methods are particularly suitable for LVLM compared to LLM, CNN, etc. It would be better that the authors can discuss more details about their insight or motivation for those designs.
- The main advantages of post-training quantization compared to quantization-aware training is that it does not need the entire training data and requires less computation. Therefore, it would be better that the authors can provide some ablations in terms of the amount of training data used during the post-training quantization and the number of computation needed during quantization. Otherwise, it is difficult to position this paper w.r.t. other relevant work.
- The authors claimed the prior quantization methods are suboptimal. However, adopting entropy as proxy and separating layers into blocks are also sub-optimal solutions. Then, there exists another question to be answered, why the sub-optimal solutions proposed in the paper are better than the others?

**Questions:**

See weaknesses

**Limitations:**

Yes

---

> ### Author Rebuttal · Authors · 2024-08-06
>
> We would like to thank the reviewer for the careful reading and valuable comments. We address the questions and clarify the issues accordingly as described below.
>
> **Q1: Confusion about the quantization strategies.**
>
> **[Reply]** We apologize for the confusion. The detailed construction of the baseline quantization methods is described in Section 4.3 as we leverage RPTQ and Outlier Suppression for activation quantization in single language and vision modality model. For our methods, the language model employs cross-layer dependency mining method exclusively, while the vision model not only utilizes cross-layer dependency mining but also incorporates additional optimization for vision encoder with vision-language synergistic optimization. Our vision encoder optimization method utilizes Jacobian matrices to assign various importance weights to different layers for vision-language synergistical optimization. We did not quantize the projector, as it contains only 20M parameters which is negligible compared to the 7B parameters of the entire model. Additionally, remaining the projector to FP would not result in significant increase in inference speed by 0.06h (6.13h vs. 6.07h) compared with 4-bit projector.
>
> **Q2: Explain why the proposed methods particularly suitable for LVLM.**
>
> **[Reply]** According to the **Overall Author Rebuttal Q2**, we point out our visual encoder optimization method is designed specifically for LVLMs due to the multi-modal data distribution compared with LLM. CNNs primarily rely on the layer-by-layer transmission of local features within the receptive field and the pooling operations that gradually lose detailed information. This makes it challenging for them to achieve global information capture and direct associations like the self-attention mechanism of LVLMs.
>
> **Q3: Ablation study about the number of training data.**
>
> **[Reply]** Thanks for your advice. We have conduct ablation studies to evaluate the effectiveness of varying amounts of calibration images used during the post-training quantization on both accuracy and calibration cost. Leverage more calibration images can reduce the rounding function overfitting, while may result in extreme distribution data selection and increased calibration time overhead with large-scale learnable parameters. The results are presented as follows:
> |Model|Calibration Images|Accuracy|Calibration Cost|
> |---|---|---|---|
> |7B|16|78.85|2.1|
> ||64|**79.79**|**8.8**|
> ||256|79.61|34.9|
> |13B|16|79.06|3.4|
> ||64|**80.78**|**14.2**|
> ||256|80.96|56.4|
>
> Observing the accuracy and calibration cost for different amounts of calibration images settings, we analyze that medium calibration images achieve the optimal performance. Limited amounts of calibration images face the challenges of overfitting, while high calibration images select extreme distribution data with marginal improvements in accuracy and substantially increased calibration cost due to large-scale learnable parameters. We suggest that utilizing 64 calibration images achieve optimal trade-off between quantization accuracy and rounding function generalizability.
>
> **Q4: Confusion about the mining cross-layer dependence.**
>
> **[Reply]** We detailly describe the significance of leveraging entropy as a proxy which identifies sensitive layers and facilitates optimal layer allocation through efficient cross-layer dependency correlation in **Rebuttal for TnuV Q2**. In data-limited PTQ quantization, it is impractical to minimize discrepancies by reconstructing the network final output to achieve the optimal solution for large-scale second-order errors. Directly searching the optimal rounding function is NP-hard because the search space increases exponentially with the layer number. Compared with conventional layer-wise quantization which minimizes quantization errors for each layer in the greedy way to solve that NP-hard problem, our method considers cross-layer dependency and employs entropy as a proxy for block-wise quantization to achieves a satisfying trade-off between discretization errors and search cost. Our cross-layer dependency mining method outperforms BRECQ by **0.56 (78.66 vs. 78.10) and 1.04 (79.89 vs. 78.85)** in the 7B and the 13B models respectively. Consequently, our method achieves optimal block partitioning and effectively utilizes cross-layer dependency.

---

> ### Comment · Area_Chair_fXuQ · 2024-08-13
>
> Dear Reviewer,
>
> Thanks for reviewing this paper! The authors have provided rebuttal to address your concerns. Could you have a look and let the authors know if you have further questions?
>
> Thanks,
> Your AC

---

### Official Review · Reviewer_TNuV · 2024-07-24

**Soundness:** 2
**Presentation:** 3
**Contribution:** 2
**Rating:** 5
**Confidence:** 4

**Summary:**

The paper introduces Q-VLM, a PTQ framework for VLMs that leverages entropy as a proxy to manage cross-layer dependencies for both language model and visual encoder. Experimental results on ScienceQA, VizWiz, VQAv2 datasets and LLaVA variant architectures validate the efficacy of the proposed method.

**Strengths:**

* Q-VLM achieves good performance with W6A6 on ScienceQA, VizWiz, VQAv2 compared to the FP counterparts.
* Ablation studies in Table 1 and Section 4.2 are helpful for understanding the importance of each component of Q-VLM, such as cross-layer dependency mining and visual encoder optimization.
* Experiments across LLaVa model sizes and bit-width demonstrates the robustness of the proposed method.

**Weaknesses:**

There are several concerns:
* The benefit and motivation for using entropy as a proxy are unclear. Why not directly use the quantization error as a metric for performing block-wise searches to mine cross-layer dependency? Are there any accuracy or inference efficiency gains by using entropy as a proxy?
* The idea of mining cross-layer dependency is not new. Existing works, such as BRECQ, already address block-wise quantization.
* Important baseline methods are missing in the comparison results. For a more solid comparison, include other state-of-the-art PTQ methods for the vision branch and more recent works such as SmoothQuant and ZeroQuant variants for the language branch. Comparing with the previous best method on the language model and another prior method on the vision model would provide a clearer picture of the overall improvement.
* The proposed method appears to be a general approach for both language and vision models. It would be interesting to demonstrate its individual effect on either the language or vision model.

**Questions:**

Suggest to clarify the concerns in the weakness section

**Limitations:**

yes

---

> ### Author Rebuttal · Authors · 2024-08-06
>
> Thank you for your valuable feedback. We appreciate the opportunity to clarify the points you raised regarding our methodology and its contributions.
>
> **Q1: The benefit and motivation for using entropy as a proxy are unclear.**
>
> **[Reply]** We are sorry for the confusion. For block-wise quantization, we partition the entire model into different blocks and search the optimal rounding function by considering the output quantization errors for each block achieves better trade-off between the search cost and the quantization accuracy. Our goal is to obtain the **optimal block partition** for rounding function search, where we leverage **Information Entropy** to judge the sensitive layers with homogeneous distribution. For sensitive layers, the noise in the former layer caused by the deviation from the global optimal value usually leads to higher deviation for the current layer, so that jointly search these layers decrease the block output quantization errors. As a result, discretization error difference (DED) between layer-wise search and joint search is obvious for sensitive layers with high entropy. The benefit and motivation for using entropy rather than quantization errors as a proxy for block-wise searches lie in several key considerations.
>
> First, we analyze that larger entropy indicates more homogeneous data distribution, which is a well-established principle in information theory. Consequently, DED and activation entropy are strongly correlated with an $R^2$ value of **0.97**. However, greater quantization error does not necessarily imply more homogeneous data distribution and does not show a positive correlation with DED, having an $R^2$ value of **0.81**, which is empirically verified in the figure presented in the Author Rebuttal.
>
> Meanwhile, the search cost of quantization errors doubles compared with entropy as a proxy, as the calculation of quantization errors requires multiple forward passes for both the FP model and the quantized model. The weak correlation and the unbearable search cost render quantization error unsuitable as a metric for measuring cross-layer dependency.
>
> Furthermore, we conducted experiments comparing the proxy effectiveness of quantization error and entropy across different models under various bitwidths. Entropy outperformed quantization errors by a significant margin **(78.66 vs. 77.97)**, showing a strong cross-layer dependency within each block. This allowed us to achieve optimal block partitioning by mining the cross-layer dependency.
> |Model|6bit Method|6bit Accuracy|6bit Search Cost|4bit Method|4bit Accuracy|4bit Search Cost|
> |---|---|---|---|---|---|---|
> |7B|Errors|88.59|43.7|Errors|77.97|41.2|
> ||Entropy|88.95|26.9|Entropy|**78.66**|**25.9**|
> |13B|Errors|88.96|58.6|Errors|78.82|54.2|
> ||Entropy|89.04|33.1|Entropy|**79.89**|**31.8**|
>
> **Q2: The idea of mining cross-layer dependency is not new.**
>
> **[Reply]** Existing works such as BRECQ incorporates Fisher information and jointly optimizes two layers within each residual block, it does not capture interactions across neighboring residual layers. Additionally, they do not provide the optimal configuration of the reconstruction granularity, with their choice of block-wise optimization stemming solely from experimental results. Consequently, these methods leverage fixed block partitions for rounding function search, leading to suboptimal performance in large models. We conduct experiments on LLaVA model with 7B and 13B parameters under 4-bit in ScienceQA dataset. Our cross-layer dependency mining method outperforms BRECQ by **0.56 (78.66 vs. 78.10) and 1.04 (79.89 vs. 78.85)** in the 7B and the 13B models respectively. Conventional quantization methods[1-3] for LLMs indicate that as model scale increases, systematic outliers with large magnitude emerge in activations, leading to significant quantization errors and accuracy degradation. Therefore, BRECQ's fine-grained block partitioning cannot effectively address outliers in LVLMs under low-bitwidth. On the country, our cross-layer dependency mining method which leverage Information Entropy achieve optimal block partition and sufficiently utilizes the cross-layer dependency.
>
> **Q3: Performance on more baseline methods.**
>
> **[Reply]** In response to **Overall Author Rebuttal Q1** as well as Q2 from your previous responds, we have conducted experiments with additional state-of-the-art PTQ methods including BRECQ and QLoRA for the vision branch and more recent methods such as ZeroQuant-V2 for the language branch. We further performed experiments combining ZeroQuant-V2 and BRECQ as baseline methods. Our Q-VLM method outperforms the baseline method by 1.71 (79.79 vs. 78.08) for answering accuracy in ScienceQA dataset under 4-bit in LLaVA-7B model. Both ZeroQuant-V2 and BRECQ are insufficient handling severe outliers in LVLM under low bitwidth which significantly degrades performance. In contrast, our Q-VLM method effectively mines cross-layer dependency by leveraging Information Entropy and utilizing Jacobian matrices to assign various importance weights to different layers for vision-language synergistical optimization.
>
> **Q4: Why the proposed methods are particularly suitable for LVLM.**
>
> **[Reply]** According to the **Overall Author Rebuttal Q2** and the ablation study in Table 1, we conducted experiments to demonstrate individual effect of cross-layer dependency mining method on both LLaMA language model and the vision model. Solely deploy cross-layer dependency mining method indicate limited performance improvement. As the visual encoder optimization method with vision-language synergistic optimization, we conclude our Q-VLM are particularly suitable for LVLM.
>
> [1] Gpt3. int8 (): 8-bit matrix multiplication for transformers at scale.
>
> [2] Smoothquant: Accurate and efficient post-training quantization for large language models.
>
> [3] Outlier suppression+: Accurate quantization of large language models by equivalent and optimal shifting and scaling.

---

> ### Comment · Area_Chair_fXuQ · 2024-08-13
>
> Dear Reviewer,
>
> Thanks for reviewing this paper! The authors have provided rebuttal to address your concerns. Could you have a look and let the authors know if you have further questions?
>
> Thanks,
> Your AC

---

> > ### Comment · Reviewer_TNuV · 2024-08-14
> >
> > Thank you to the authors for the detailed response. Most of my concerns have been well-addressed, so I am raising my rating to 5: Borderline Accept. However, I still believe it would be valuable to provide a more solid comparison by benchmarking the proposed method against a baseline that includes the best prior method for the language branch and another existing top approach for the vision branch. The baseline method does not have to be the same for both the language and vision parts.

---

> > > ### Author Response · Authors · 2024-08-14
> > >
> > > We are truly grateful for your valuable feedback on our paper. We provide a more solid comparison by benchmarking the proposed method against a baseline that includes ZeroQuant-V2 for the language branch and BRECQ for the vision branch. Our Q-VLM method outperforms the baseline method by 1.71 (79.79 vs. 78.08) in ScienceQA dataset under 4-bit in LLaVA-7B model. Both ZeroQuant-V2 and BRECQ fail to handle severe outliers in LVLM under low bitwidth which significantly degrades performance. In contrast, our Q-VLM method effectively mines cross-layer dependency and achieves vision-language synergistical optimization. Due to time constraints, we plan to implement SOTA DopQ-ViT [1] instead of BRECQ for the vision branch in future work.
> > >
> > > [1] Yang, Lianwei, and Haisong Gong. "DopQ-ViT: Towards Distribution-Friendly and Outlier-Aware Post-Training Quantization for Vision Transformers." arXiv preprint arXiv:2408.03291 (2024).

---

### Author Rebuttal · Authors · 2024-08-06

We appreciate the valuable feedback and insightful questions provided by the reviewers. Below are our detailed responses to two common concerns raised by multiple reviewers. Detailed responses to other specific comments are provided under each individual reviewer's comments.

**Q1: Performance on more baseline methods and different multi-modal architectures.**

**[Reply]** Thanks for suggestions which would provide a deeper insight into the effectiveness and generalizability of our method. We have extended our experiments to an additional baseline method ZeroQuant-V2[1] and compared it against our proposed methods. ZeroQuant-V2 leverages per-token quantization with different rounding functions to minimizing activation discretization errors. However, ignoring cross-layer dependency of discretization errors fails to acquire the optimal rounding strategy with severe outliers under low bitwidth and degrades the performance significantly. On the contrary, our Q-VLM mines the cross-layer dependency of output distribution across layers, minimizing the block-wise discretization errors to avoid suboptimal quantization. We further optimize the visual encoder to disentangle the cross-layer dependency for fine-grained search space decomposition. As a result, our method outperforms ZeroQuant-V2 by **1.71 (79.79 vs. 78.08)** in answering accuracy on ScienceQA dataset under 4-bit in LLaVA-7B model. Additionally, our method enhances inference speed, exceeding ZeroQuant-V2 by **1.2h (6.1h vs. 7.3h)** due to utilizing stored rounding parameters instead of dynamic per-token quantization. The results are presented as follows:
|Model|8bit Method|8bit Accuracy|4bit Method|4bit Accuracy|
|---|---|---|---|---|
|7B|ZeroQuant-V2 [1]|89.04|ZeroQuant-V2 [1]|78.08|
||Q-VLM|**89.58**|Q-VLM|**79.79**|
|13B|ZeroQuant-V2 [1]|89.13|ZeroQuant-V2 [1]|78.81|
||Q-VLM|**89.81**|Q-VLM|**80.78**|

The additional baseline provides a more comprehensive evaluation framework to highlight the strengths of our approach. We also explored the multi-modal architecture OpenFlamingo [2] to ensure the robustness and generalizability of our methods. We deploy our method on OpenFlamingo 3B model using Vizwiz and Hateful Memes[3] datasets, selecting bitwidths of 4 and 8 for quantized layers. The experiments are presented as follows:
|Dataset|Shots|FP|8bit Q-LoRA|8bit Q-VLM|4bit Q-LoRA|4bit Q-VLM|
|---|---|---|---|---|---|---|
|Vizwiz|0|23.79|21.24|21.47|17.62|18.69|
||4|27.05|25.83|26.59|24.17|24.55|
||32|39.76|36.38|**37.60**|31.64|**35.52**|
|Hateful Memes [3]|0|50.23|47.75|49.12|43.86|44.22|
||4|50.10|48.62|49.55|45.12|45.26|
||32|50.27|50.02|**51.05**|45.76|**47.84**|

Q-VLM designed in LLaVA-like architectures can be effectively adapted to cross-attention based VLMs due to the consistent core mechanism of cross-attention and the robust multimodal alignment capabilities pre-trained on large-scale vision-language pairs. Since OpenFlamingo is a cross-attention based VLM, exploiting cross-layer dependency is particularly suitable. Our method outperforms Q-LoRA by **1.22 (37.60 vs. 36.38)** under 8-bit in OpenFlamingo-3B model. The advantage of our method becomes more obvious for 4-bit 3B LVLMs because quantization errors and cross-layer dependency play a more significant role in networks with low capacity. These results underscore the robustness and generalizability of our approach across different tasks, model architectures and datasets, demonstrating its effectiveness in diverse scenarios.

[1] Yao, Zhewei, et al. "Zeroquant-v2: Exploring post-training quantization in llms from comprehensive study to low rank compensation." arXiv preprint arXiv:2303.08302 (2023).

[2] Awadalla, Anas, et al. "Openflamingo: An open-source framework for training large autoregressive vision-language models." arXiv preprint arXiv:2308.01390 (2023).

[3] Kiela, Douwe, et al. "The hateful memes challenge: Detecting hate speech in multimodal memes." Advances in neural information processing systems 33 (2020): 2611-2624.

**Q2: Why are the proposed methods particularly suitable for LVLM?**

**[Reply]** We appreciate the reviewer's interest in understanding the suitability of our proposed methods for LVLM. Our cross-layer dependency mining method is also suitable for single modality model. Specifically, utilizing entropy as proxy to assess layer sensitivity and achieve optimal block partitioning is not exclusively beneficial for LVLMs. For single language modality, we deploy our CDM into the LLaMA-7B[1] model and evaluated it on the PIQA[2] dataset. Our method achieved improvement compared to QLoRA by **0.51 (72.23 vs. 71.82)** in 4-bit precision. For single vision modality, we incorporated our CDM into the CLIP only quantized LLaVA-7B model and observed enhancement of about 0.23 (84.15 vs. 83.92) in answering accuracy on the ScienceQA dataset under 4-bit precision.

However, ablation studies in Table 1 indicate limited performance improvement by 1.13 (78.66 vs. 77.53) when solely using cross-layer dependency mining without visual encoder optimization method. The proposed visual encoder optimization method is unsuitable for single-modal models from the perspective of vision-language synergistic optimization to achieve fine-grained blocks with search cost and cross-layer dependency trade-off. Significant performance gains by **2.26 (79.79 vs. 77.53)** are achieved by jointly employing cross-layer dependency mining and visual encoder optimization method. Our approach achieves vision-language synergistic optimization, resulting in superior performance in LVLM tasks.

[1] Touvron, Hugo, et al. "Llama: Open and efficient foundation language models." arXiv preprint arXiv:2302.13971 (2023).

[2] Bisk, Yonatan, et al. "Piqa: Reasoning about physical commonsense in natural language." Proceedings of the AAAI conference on artificial intelligence. Vol. 34. No. 05. 2020.

---

### Decision · Program_Chairs · 2024-09-25

**Decision:**

Accept (poster)

**Comment:**

Previous post-training quantization is to search the layer-wise rounding functions sequentially. This paper claims it is suboptimal, and proposes to leverage some cross-layer dependency and search block-wise rounding functions. It also proposes to use entropy as the proxy to partition the blocks optimally. The results have shown no much performance regression when quantizing LLava model.

Most of the reviews stay positive on this paper due to 1) it is reasonable and well motivated to search block-wise rounding functions; 2) the results has very minor performance regression. But he common concerns are: 1) the motivation to use entropy as proxy; 2) generalization to single-modality models; 3) missing more evaluation on other benchmarks; 4) more fair comparison with baselines. After the rebuttal, most of the concerns are resolved, but some are still remained, e.g. missing stronger baselines, the method is not specifically for VLM but also applicable to single-modality models. The authors should reflect the rebuttal and further improve the remaining issues in the final version.